# Neural population dynamics in human motor cortex during movements in people with ALS

Chethan Pandarinath[1,2,3], Vikash Gilja[1,2,4], Christine H Blabe[1], Paul Nuyujukian[1,2,3], Anish A Sarma[4,5,6,7], Brittany L Sorice[6], Emad N Eskandar[8,9], Leigh R Hochberg[5,4,6,7,10], Jaimie M Henderson[1,3†], Krishna V Shenoy[2,3,11,12,13]*†

[1]Department of Neurosurgery, Stanford University, Stanford, United States; [2]Department of Electrical Engineering, Stanford University, Stanford, United States; [3]Stanford Neurosciences Institute, Stanford University, Stanford, United States; [4]School of Engineering, Brown University, Providence, United States; [5]Center for Neurorestoration and Neurotechnology, Rehabilitation R and D Service, Department of VA Medical Center, Providence, United States; [6]Neurology, Massachusetts General Hospital, Boston, United States; [7]Institute for Brain Science, Brown University, Providence, United States; [8]Department of Neurosurgery, Harvard Medical School, Boston, United States; [9]Department of Neurosurgery, Massachusetts General Hospital, Boston, United States; [10]Neurology, Harvard Medical School, Boston, United States; [11]Department of Neurobiology, Stanford University, Stanford, United States; [12]Department of Bioengineering, Stanford University, Stanford, United States; [13]Neurosciences Program, Stanford University, Stanford, United States

*For correspondence: shenoy@stanford.edu

†These authors contributed equally to this work

Competing interests: The authors declare that no competing interests exist.

**Abstract** The prevailing view of motor cortex holds that motor cortical neural activity represents muscle or movement parameters. However, recent studies in non-human primates have shown that neural activity does not simply represent muscle or movement parameters; instead, its temporal structure is well-described by a dynamical system where activity during movement evolves lawfully from an initial pre-movement state. In this study, we analyze neuronal ensemble activity in motor cortex in two clinical trial participants diagnosed with Amyotrophic Lateral Sclerosis (ALS). We find that activity in human motor cortex has similar dynamical structure to that of non-human primates, indicating that human motor cortex contains a similar underlying dynamical system for movement generation.
Clinical trial registration: NCT00912041.

## Introduction

Neurons in motor cortex exhibit complex firing patterns during movement (*Fetz, 1992*; *Churchland and Shenoy, 2007*; *Churchland et al., 2010*). Though motor cortex has been extensively studied over the past century (*Lemon, 2008*), the complexity of these patterns remains poorly understood. Early work demonstrated that neural firing correlates with external variables such as movement direction (*Georgopoulos et al., 1982*), and an ongoing debate centers on whether the firing patterns represent muscle or movement parameters (e.g., joint position, intended velocity, reach endpoint, muscle forces, and so on; reviewed in *Kalaska, 2009*). A recent approach has been to ask, instead, whether these patterns of activity reflect an internal dynamical system across the population (*Fetz, 1992*; *Todorov and*

**eLife digest** Every conscious movement a person makes, whether lifting a pencil or playing a violin, begins in the brain. To be more specific, neurons in a part of the brain called the motor cortex send signals to muscles to cause them to move.

But many of the details about how messages from the motor cortex produce movements remain unclear. Some scientists believe that individual neurons in motor cortex send direct messages that tell the muscles which direction to move in, how fast, and how forcefully. But other scientists suggest that this is not the case. Instead, they propose that neurons in motor cortex work together as part of a dynamic system to create rhythmic patterns of activity for movement. These rhythmic patterns then sum together to create the signals that muscles need to carry out the movements.

Studies in monkeys have supported the idea that the neurons in the motor cortex are not just direct messengers. These studies showed what appears to be a rotating set of patterns in neuronal activity in the motor cortex during movement. Now, Pandarinath et al. have shown that a similar rotation of neuronal activity patterns occurs during movement in two human volunteers. The participants both had a disease called amyotrophic lateral sclerosis (or ALS for short). This disease had nearly paralyzed their arms and legs because it causes a progressive loss of muscle control. In the experiments, the volunteers used their index fingers to try to move a computer cursor to a target using a touch pad.

Pandarinath et al. recorded activity in the volunteers' motor cortices while they completed the tasks. The experiments uncovered a predictable series of patterns that started when the individual first thought about moving. These patterns progressed in a rotation as the movement was carried out. The rotation was not tied to the direction of the movements and would be completely unexpected if the individual neurons were simply acting as direct messengers.

It is hoped that these findings will help efforts to create prosthetic devices (such as robotic arms) that can better respond to an individual's thoughts. But further experiments are also needed in people without ALS to verify that the patterns observed weren't specifically related to this disease.

---

*Jordan, 2002*; *Churchland and Shenoy, 2007*; *Scott, 2008*; *Churchland et al., 2010*, *2012*; *Graziano, 2011*; *Shenoy et al., 2013*; *Kaufman et al., 2014*). In this view, the system's initial state is set by preparatory activity, and consistent rules govern how firing rates evolve over time across movement conditions, regardless of movement direction or other externally changing variables.

This dynamical view is supported by recent studies in rhesus macaques (*Churchland et al., 2010*, *2012*; *Shenoy et al., 2013*; *Kaufman et al., 2014*). At the level of single neurons, responses during movement show brief but strong oscillatory components, even for straight point-to-point reaches (*Churchland and Shenoy, 2007*; *Churchland et al., 2010*). Furthermore, at the population level, the responses are well-described by a simple dynamical model in which the population response (the neural state) rotates with time (*Figure 1A*) (*Churchland et al., 2012*). Importantly, the presence of a strong rotational component in these dynamics is not predicted by the prevailing view of motor cortical activity (i.e., cosine direction tuning, linear speed scaling) that, instead, predicts solely expansive and contractive dynamics (detailed in *Churchland et al., 2012*, Figure 4 and associated text).

In humans, studies in research participants with tetraplegia have demonstrated that motor cortical action potential (AP) responses contain information about intended movement kinematics (*Truccolo et al., 2008*). Here, we investigated whether these AP responses also contain rotational dynamical structure at the ensemble level.

## Results

We analyzed multi-neuron AP activity from 2 people with tetraplegia enrolled in the BrainGate2 pilot clinical trial. This ongoing, multi-site, pilot clinical trial is performed under an FDA Investigational Device Exemption and has been approved by local institutional review boards at all study sites. The study participants (T6, T7) had differing degrees of motor impairment due to Amyotrophic Lateral Sclerosis (ALS). T6 retained the ability to make several dexterous movements (especially of the fingers and wrist), while T7 retained limited finger movements.

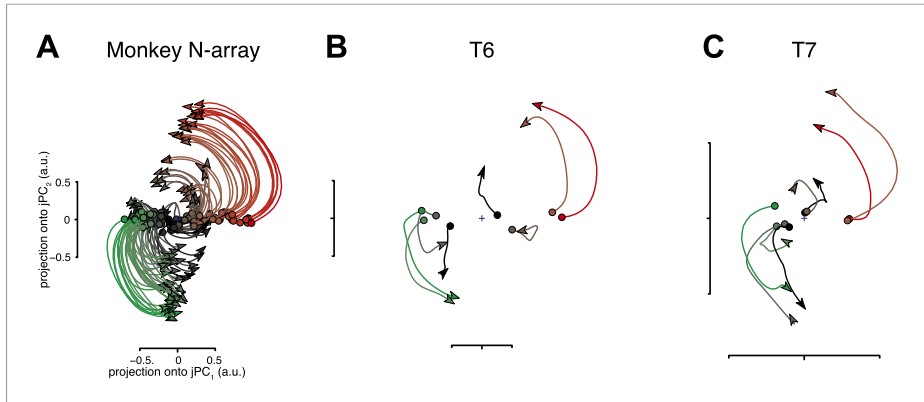

**Figure 1**. Neural population responses show rotational activity during movement epochs. (**A**) Projections of the neural population response onto the first jPCA plane for a monkey during an arm-reaching task (monkey N, 108 conditions; adapted from *Churchland et al., 2012*). Each trace plots the first 200 ms of activity during the movement epoch for a given condition. Traces are colored based on the preparatory state projection onto $jPC_1$. a.u., arbitrary units. (**B**) Projections for participant T6 during an 8-target center-out task controlled by index finger movements on a computer touchpad. Each trace plots the 250 ms of activity during the movement epoch ('Materials and methods') for a given condition. (**C**) Same as (**B**), for participant T7. *Video 1* shows the evolution of the neural state over time for each participant.

Neural signals were recorded using 4 mm × 4 mm, 96-channel silicon microelectrode arrays, which were implanted in the hand area of dominant M1. APs were recorded as participants performed visual target acquisition tasks. Participants attempted to move a cursor from the center of a computer screen to one of eight peripheral targets, with the cursor's position controlled by index finger movements on a computer touchpad (see 'Materials and methods').

We first tested whether an underlying dynamical structure existed in the neural activity. If present, a population-level analysis should reveal orderly rotational structure that is consistent across conditions (i.e., independent of the direction of movement). To search for rotational patterns in the neural population state, we used a three-step procedure: first, for each condition (i.e., for a given target), we averaged the activity on each electrode across all trials. Next, we performed principal components analysis (PCA) on the high-dimensional population data. We restricted the data to the top 6 PCs, that is, we only preserved the six response patterns most strongly present in the data. Finally, for this reduced-dimensional data set, we applied the jPCA method (*Churchland et al., 2012*), which searches the data for 2-dimensional planes that capture the strongest rotational tendencies. Restricting the jPCA analysis to the dimensionality-reduced data (6-D) ensured that any rotational structure revealed by the analysis was present in the most prominent response patterns in the data.

For both participants, the population activity exhibited strong rotational dynamics (*Figure 1B,C*). Each trace shows the population activity in the top jPC plane for a single condition. 250 ms of data are shown, beginning with the rapid change in neural activity that precedes movement onset (the evolution of the neural state over time for each participant is shown in *Video 1*). Rotations proceeded in the same direction across conditions, following from the initial pre-movement state. The top jPCA plane captured 61% (T6) and 27% (T7) of the variance of the high-dimensional neural data (for comparison, the macaque study reported 28% for the top plane on average).

One potential concern is that the jPCA method might be powerful enough to find rotatory patterns in state space for any set of responses that contains complex, multiphasic patterns. To test for this possibility, we performed three control analyses, following *Churchland et al., 2012*. In these controls, the data were shuffled to disrupt underlying rotational structure across response patterns, while preserving the complexity of the individual response patterns. If the previously found rotational structure were simply a by-product of the analysis technique, then the shuffled data sets should still show prominent rotations in the top jPCA planes. This was not the case. Rotations were no longer qualitatively seen in the projected responses after shuffling (*Figure 2*, top row). We next measured the fraction of variance of the changes in neural state (6-D) that could be explained by rotational activity

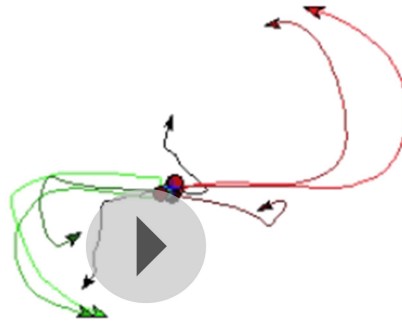

400 ms

**Video 1.** Neural population responses show rotational activity. Video shows the evolution of the neural state over time in the first jPCA plane for participants T6 and T7. Low-dimensional projections were calculated as in *Figure 1*. Each colored trace represents one of 8 conditions. All times are relative to target onset (0 ms). DOI: 10.7554/eLife.07436.004

alone (see 'Materials and methods') and found that this greatly decreased after shuffling (*Figure 2*, bottom row). (Two additional shuffle control analyses are presented in *Figure 2—figure supplement 1* and *Figure 2—figure supplement 2*).

To quantify the consistency of the rotatory activity, we measured the angle from the neural state in the jPCA plane, $x$, to its derivative, $\dot{x}$, for each time point across all conditions (*Figure 3*). Angles near pi/2 indicate rotational dynamics. As shown, the distribution of measured angles peaked near pi/2, similar to previously reported macaque data.

## Discussion

These results demonstrate prominent rotations of the neural population state in human motor cortex during movements. As with macaques, rotations were consistent across conditions and followed naturally from an initial pre-movement state. As mentioned above, and in contrast to the macaque study, both participants in this study had a diagnosis of ALS with resultant motor impairment. Participant T7's movements in particular were very limited and occurred with a long latency following target onset (detailed in 'Materials and methods'). Both participants likely had substantial changes in motor cortex due to their disease progression; therefore, these results do not conclusively show that dynamical activity is present in healthy human motor cortex. However, given the finding that dynamical activity is prominent in motor cortex in people with abnormal motor function, and given the similarity of this activity to that of healthy non-human primates, the results strongly suggest the presence of dynamical activity in human motor cortex in general and may also hint at which aspects of motor cortical function are preserved despite the progression of a severe motor neuron disease.

Given the findings of dynamical activity during overt movements, a potentially exciting open question is whether motor cortex exhibits dynamical activity during purely imagined movements. Recent studies (Feldman et al., Society for Neuroscience, 2011, 2012; Pandarinath et al., Society for Neuroscience, 2013) have demonstrated that motor cortex is active during both overt and imagined movements. If the functional role of rotational dynamics is to serve as an oscillatory basis set for generating muscle activation patterns (e.g. electromyogram activity) (*Churchland et al., 2012*; *Sussillo et al., 2015*), then one might expect these dynamics to be absent during imagined movements, when the subject is specifically trying to avoid generating motor output. However, if activity during imagined movements serves as a mechanism for the 'covert rehearsal' of activity during overt movements, then dynamics might still be present (but might be, for example, orthogonal to patterns of activity that drive motor output). Future studies will attempt to directly address this question.

The current study's findings also suggest a promising avenue to improve the performance of Brain–Machine Interfaces (BMIs) for persons with tetraplegia, as previous work with macaques (*Kao et al., 2015*) demonstrated that incorporating neural population dynamics into BMI control algorithms may lead to performance improvements. Finally, as with macaques, the presence of these rotations calls into question the prevailing model of motor cortical activity (i.e., that motor cortical firing patterns represent muscle or movement parameters) in favor of a dynamical systems perspective in humans as well.

## Materials and methods

Permission for these studies was granted by the US Food and Drug Administration (Investigational Device Exemption) and Institutional Review Boards of Stanford University (protocol # 20804), Partners

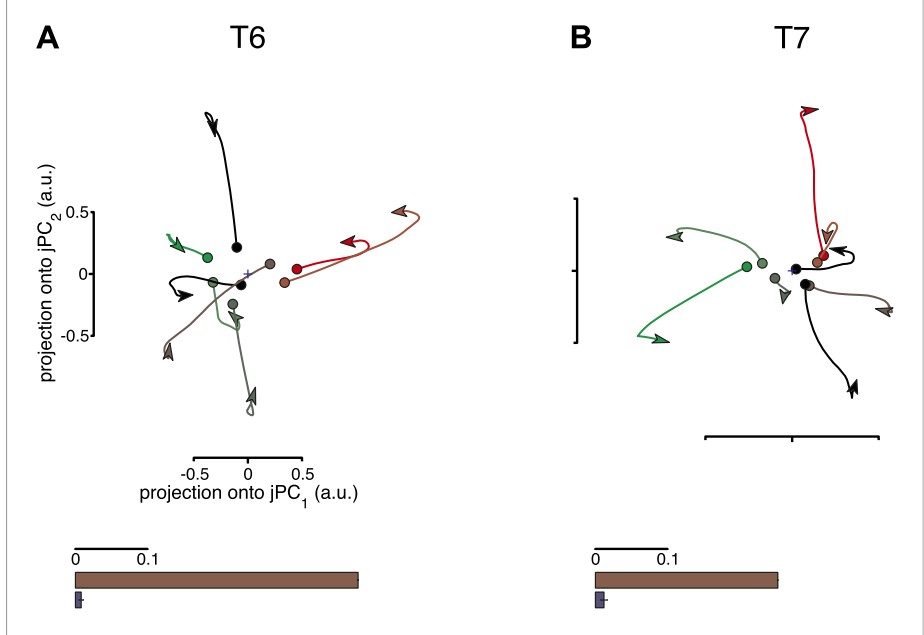

**Figure 2**. Rotational dynamics are not a by-product of the jPCA analysis method. For each data set, neural responses were shuffled in a manner that preserved the complexity of individual response patterns on each electrode, but disrupted the structure of the data across electrodes. For each channel, the pattern of activity during the movement epoch was inverted for half the conditions (chosen at random). The inversion was performed around the initial time point, so that continuity with pre-movement activity was preserved. Performing jPCA on the shuffled responses did not reveal consistent rotational structure. (**A**, *top*) Projection of the population responses onto the first jPCA plane for a single shuffled trial (participant T6). (*bottom*) Fraction of variance of the change in neural state (6-D) explained by rotational activity for the original data set (brown) vs the shuffled data sets (blue). Error bar represents the standard deviation across 300 shuffle trials. (**B**) Same as (**A**), for participant T7. Two additional shuffle control analyses are presented in *Figure 2—figure supplement 1* and *Figure 2—figure supplement 2*.

The following figure supplements are available for figure 2:

**Figure supplement 1**. Results of the second shuffle control analysis.

**Figure supplement 2**. Results of the third shuffle control analysis.

Healthcare/Massachusetts General Hospital (2011P001036), Providence VA Medical Center (2011-009), and Brown University (0809992560). The two participants in this study, T6 and T7, were enrolled in a pilot clinical trial of the BrainGate Neural Interface System (http://www.clinicaltrials.gov/ct2/show/NCT00912041). Informed consent, including consent to publish, was obtained from the participants prior to their enrollment in the study.

## Participants

Participant T6 is a right-handed woman, 51 year old at the time of this study, with tetraplegia due to ALS. On December 7, 2012, a 96-channel intracortical silicon microelectrode array (1.0-mm electrode length, Blackrock Microsystems, Salt Lake City, UT) was implanted in the hand area of dominant motor cortex as previously described (*Hochberg et al., 2006*; *Simeral et al., 2011*). T6 retained dexterous movements of the fingers and wrist. Data reported in this study are from T6's trial days 95–213.

Participant T7 is a right-handed man, 54 year old at the time of this study, with tetraplegia due to ALS. T7 had two 96-channel intracortical silicon microelectrode arrays (1.5-mm electrode length, Blackrock Microsystems, Salt Lake City, UT) implanted in the hand area of dominant motor cortex on July 30, 2013. Data reported are from T7's trial days 231 and 245. At that time, T7 retained very

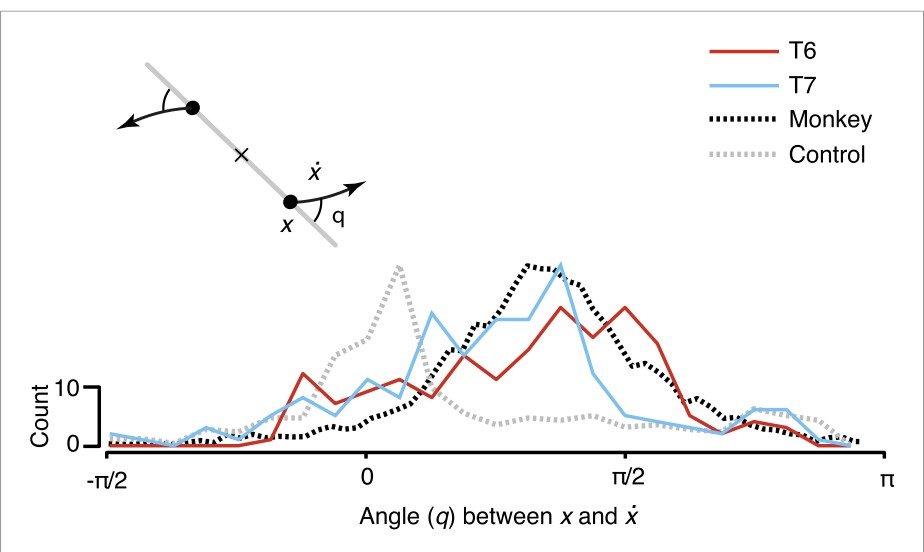

**Figure 3**. Consistency of the rotational dynamics across conditions. Traces represent histograms of the angle, $q$, between the neural state, $x$, and its derivative, $\dot{x}$, for each time step. The angle was measured as illustrated schematically (inset) after projecting the data into the first jPCA plane. Purely rotatory activity results in angles near pi/2, while pure scaling/expansion results in angles near 0 or pi. Y-axis denotes scale for participant data (colored traces). For comparison, histograms for the example shuffle control data (**Figure 2**) and monkey composite data are shown in gray and black, respectively. These traces are normalized to match the participant data range (monkey data reproduced from **Churchland et al., 2012**).

limited, but consistent, index finger movements. Subsequent to these days, further motor impairment precluded tasks that relied on finger movements.

## Task design

Neural data were recorded during 'center-out' target acquisition tasks. The data were originally collected for neural prosthetic decoder calibration, as part of research testing algorithms for closed-loop neural cursor control (Gilja et al., Society for Neuroscience, 2013). In the 'center-out' task, participants controlled the position of a cursor on a computer screen by making physical movements with their fingers on a wireless touchpad (Magic Trackpad; Apple, Cupertino, CA). The cursor began in the center of the screen, and targets would appear in one of 8 locations on the periphery. Participants then acquired the targets by moving the cursor over the target and holding it over the target for 500 ms. In contrast to the prior macaque study (**Churchland et al., 2012**), the target acquisition task used in this work did not include a delay period. Therefore, participants were free to move as soon as the target appeared. Participant T6 was not limited in her ability to span the workspace of the touchpad. Participant T7's limited movements spanned a small region on the touchpad, approximately 1/8″–1/4″ wide.

## Recordings and data analysis

Data were aggregated over multiple sessions (T6: 8 sessions, T7: 2 sessions). Trial counts varied between sessions and across participants (T6: 75–220 trials per session, T7: 128 and 78 trials per session). The primary data analyzed were multi-neuron APs, which were taken as time points when a given channel's voltage exceeded a fixed threshold. Choice of threshold was dependent on the array (T6: −60 µV, T7, Lateral array: −80 µV, Medial array: −95 µV). Analyses were restricted to electrodes known to have significant modulation during attempted movements (T6: 39 electrodes, T7: 78 electrodes). Firing rates per electrode were then averaged across trials and filtered with a Gaussian kernel with standard deviation of 25 ms (T6) or 30 ms (T7).

## Analysis of rotational structure in the population response

jPCA analyses were performed as described previously (*Churchland et al., 2012*). Analyses were restricted to a 250-ms time period beginning with the rapid changes in neural activity that occur preceding movement onset. This period began approximately 180 ms and 300 ms after target onset for T6 and T7, respectively, corresponding to a difference in reaction times between participants. Overt movement was detectable at approximately 230 ms and 600 ms after target onset for T6 and T7, respectively. The delayed movements in T7 relative to T6 likely reflect a difference in disease progression between the two participants.

Pre-processing steps ('soft' normalization, mean-centering, and initial dimensionality reduction using PCA) were performed following *Churchland et al., 2012*. The initial dimensionality reduction step restricted the data to the top 6 PCs.

jPCA is a method for finding projections that capture rotational structure in a data set. (The method is described in detail in *Churchland et al., 2012*; here, we summarize its key features.) The method is based on comparing the neural state at a given point in time with its derivative. The initial dimensionality-reduction step (performed on the trial-averaged firing rates) reduces the data to the matrix $X_{red}$, which has dimensions $d \times ct$ (where $d$ is the number of PCs kept, $c$ is the number of conditions, and t is the number of time points). We computed $\dot{X}_{red}$, of size $d \times c\,(t-1)$, by taking the difference in state between adjacent time points (the final time point of $X_{red}$ was subsequently removed to equalize the sizes of $X_{red}$ and $\dot{X}_{red}$). We then fit the neural state transition matrices, $M$ and $M_{skew}$:

$$\dot{X}_{red} = MX_{red},$$

and

$$\dot{X}_{red} = M_{skew}X_{red},$$

where $M$ is unconstrained (fit via linear regression), and $M_{skew}$ is constrained to be a skew-symmetric matrix (i.e., $M_{skew} = -M_{skew}T$, which has purely imaginary eigenvalues), and thus, captures only rotational dynamics. The first jPCA plane is then constructed from the eigenvectors of $M_{skew}$ associated with the largest eigenvalues. For a given jPCA plane, the basis vectors $jPC_1$ and $jPC_2$ are selected such that the pre-movement activity (the initial state) is maximally spread along $jPC_1$, and that the net rotation in the plane is anticlockwise.

To measure the fraction of variance of the changes in neural state (i.e., the dimensionality-reduced data) that could be explained by rotational activity (*Figure 3*, bottom panels), the data were modeled as a linear dynamical system, where $M_{skew}$ perfectly captured the dynamics between time $t$ and $t-1$, that is, only purely rotational dynamics were allowed. Fraction of state variance explained for real and shuffled data was estimated using the top 6 PCs.

## Shuffle control analyses

One potential concern is that the jPCA method might be powerful enough to find rotatory patterns in state-space for any set of responses that contains complex, multiphasic patterns. The likelihood of finding such spurious rotatory patterns, which are common across conditions, increases as the number of conditions decreases. Thus, this is a larger concern in the current work (8 conditions per participant) than previous work with macaques (27–108 conditions per monkey) (*Churchland et al., 2012*).

To test for this possibility, we performed the three control analyses performed in *Churchland et al., 2012* (described in their Supplementary Figures 2 and 3). Each of the three 'shuffle' controls preserves the diversity and complexity of responses, but perturbs the structure of the responses at the population level. Specifically, the dynamical model assumes that neural activity during the movement epoch follows in an orderly fashion from its pre-movement state. To test this assumption, the three shuffle control analyses disrupt the relationship between the pre-movement activity and the movement epoch for each channel. If these shuffled data sets still showed prominent rotatory activity, it would indicate that rotations might be found by the jPCA method even when not truly present.

## Acknowledgements

The authors would like to thank participants T6, T7, and their families; B Davis, B Pedrick, E Castenada, M Coburn, B Travers, and D Rosler for administrative support; L Barefoot, P Gigante, A Sachs, S Cash, J Menon, SI Ryu, and S Mernoff for clinical and surgical assistance; J Saab and N Schmansky for technical assistance; JP Cunningham, MM Churchland for helpful comments on the manuscript and jPCA and related MATLAB code; MT Kaufman for helpful comments on the manuscript; JP Donoghue for helpful scientific discussions.

This work was supported by: Stanford Institute for Neuro-Innovation and Translational Neuroscience; Stanford BioX/NeuroVentures; Stanford Office of Postdoctoral Affairs; Garlick Foundation; Craig H. Neilsen Foundation; National Institutes of Health: NIDCD (R01DC009899), NICHD-NCMRR (N01HD53403 and N01HD10018), NINDS (R01NS066311); Rehabilitation Research and Development Service, Department of Veterans Affairs (B6453R and B6310N); MGH-Deane Institute for Integrated Research on Atrial Fibrillation and Stroke.

The content is solely the responsibility of the authors and does not necessarily represent the official views of the National Institutes of Health, or the Department of Veterans Affairs or the United States Government. CAUTION: Investigational Device. Limited by Federal Law to Investigational Use.

## Additional information

### Funding

| Funder | Grant reference | Author |
| --- | --- | --- |
| National Institutes of Health (NIH) | R01DC009899 | Leigh R Hochberg |
| National Institutes of Health (NIH) | N01HD53403 | Leigh R Hochberg |
| National Institutes of Health (NIH) | N01HD10018 | Leigh R Hochberg |
| National Institutes of Health (NIH) | R01-NS066311 | Jaimie M Henderson, Krishna V Shenoy |
| U.S. Department of Veterans Affairs | B6453R, B6310N | Leigh R Hochberg |
| Garlick Foundation | | Jaimie M Henderson, Krishna V Shenoy |
| Craig H. Neilsen Foundation | | Chethan Pandarinath |
| Massachusetts General Hospital | Deane Institute for Integrated Research on Atrial Fibrillation and Stroke | Leigh R Hochberg |
| Stanford University | Institute for Neuro-Innovation and Translational Neuroscience | Jaimie M Henderson, Krishna V Shenoy |
| Stanford University | Stanford BioX/ Neuroventures | Jaimie M Henderson, Krishna V Shenoy |
| Stanford University | Stanford Office of Postdoctoral Affairs | Chethan Pandarinath |

The funders had no role in study design, data collection and interpretation, or the decision to submit the work for publication.

### Author contributions

CP, JMH, KVS, Conception and design, Acquisition of data, Analysis and interpretation of data, Drafting or revising the article; VG, CHB, PN, AAS, BLS, ENE, LRH, Acquisition of data, Drafting or revising the article

## Author ORCIDs

Chethan Pandarinath, http://orcid.org/0000-0003-1241-1432
Paul Nuyujukian, http://orcid.org/0000-0001-7778-5473
Anish A Sarma, http://orcid.org/0000-0003-1261-0589
Leigh R Hochberg, http://orcid.org/0000-0003-0261-2273
Jaimie M Henderson, http://orcid.org/0000-0002-3276-2267
Krishna V Shenoy, http://orcid.org/0000-0003-1534-9240

## Ethics

Clinical trial Registry: NCT. Registration ID: NCT00912041.

Human subjects: Permission for these studies was granted by the US Food and Drug Administration (Investigational Device Exemption) and Institutional Review Boards of Stanford University (protocol # 20804), Partners Healthcare/Massachusetts General Hospital (2011P001036), Providence VA Medical Center (2011-009), and Brown University (0809992560). The two participants in this study, T6 and T7, were enrolled in a pilot clinical trial of the BrainGate Neural Interface System (http://www.clinicaltrials.gov/ct2/show/NCT00912041). Informed consent, including consent to publish, was obtained from the participants prior to their enrollment in the study.

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
