## [Decision Letter]

Thank you for sending your work entitled “Neural population dynamics in human motor cortex during movements” for consideration at *eLife*. Your article has been favorably evaluated by Eve Marder (Senior editor), a Reviewing editor, and three reviewers.

The following individuals responsible for the peer review of your submission have agreed to reveal their identity: Ole Kiehn (Reviewing editor), Michael Graziano and Timothy Ebner (peer reviewers).

Summary:

This manuscript by Pandarinath et al. out of the Shenoy laboratory used single cell recordings from the primary motor cortex in patients with tetraplegia due to ALS. The recordings were from chronically implanted electrode arrays used for brain machine interface studies to restore function. The study tests a new dynamical systems view of motor cortex in humans, as it has previously been tested in non-human primates (NHPs). The study confirms the view and shows that the work in NHPs applies to humans.

Essential Revisions:

The Reviewing editor and the reviewers discussed their comments before we reached this decision, and the Reviewing editor has assembled the following comments to help you prepare a revised submission.

1) The methodology and analysis is sound and overall the reviewers were very positive about the study and find that the importance of the study is that it extends the work in NHPs to humans. Without being a requirement for new experiment it is noted that the investigators could have taken the opportunity to investigate motor activity that can be done in humans but not in NHPs, for example purely imaginary movements.

2) Some words about to what extent the recorded activity can be extrapolated to normal motor cortex function is needed. ALS is a severe motor neuron disease. The Title and Abstract should at least be modified to reflect this (Neural population dynamics in human motor cortex during movements in patients with ALS) and mentioning of this in the Discussion is needed. The fact that the dynamical structure was found may speak to the robustness of the observations.

3) It will be appropriate to address possible ethical issues of using invasive surgical experiments in patients with a severe disease and limited motor function.

---

## [Author Response]

*1) The methodology and analysis is sound and overall the reviewers were very positive about the study and find that the importance of the study is that it extends the work in NHPs to humans. Without being a requirement for new experiment it is noted that the investigators could have taken the opportunity to investigate motor activity that can be done in humans but not in NHPs, for example purely imaginary movements*.

This is a very insightful point, and a promising avenue for future research. We agree, understanding whether ensemble activity exhibits rotational dynamics during imagined movements is a very interesting scientific question. More broadly, we are interested in discerning the cognitive and neurophysiological correlates of imagined or intended movements in people with tetraplegia, and are pursuing this topic as part of our ongoing research. However, we feel this is beyond the scope of the current work. We believe, and the reviewers seem to agree, that the importance of the current short report is the clinical translation of a key finding from non-human primate studies. In contrast, we don't have a good prediction, based on the prior work, for whether motor cortex should show dynamical activity during imagined movements.

Concretely, if the functional role of rotational dynamics is to serve as an oscillatory basis set for EMG generation ([2], Sussillo et al., 2015), then one might expect these dynamics to be absent during imagined movements, when the subject is specifically trying not to move. However, if activity during imagined movements serves as a mechanism for the “covert rehearsal” of activity during overt movements, then dynamics might still be present (but might be, for example, orthogonal to patterns of activity that drive motor output). Due to the complexity of this issue, and the necessity for dedicated experiments and controls, we feel this is better addressed in a follow-on study than in this short report.

We agree that it is beneficial to raise this interesting question, and have added text (similar to the above) to the Discussion to highlight this issue.

*2) Some words about to what extent the recorded activity can be extrapolated to normal motor cortex function is needed. ALS is a severe motor neuron disease. The Title and Abstract should at least be modified to reflect this (Neural population dynamics in human motor cortex during movements in patients with ALS) and mentioning of this in the Discussion is needed. The fact that the dynamical structure was found may speak to the robustness of the observations*.

We agree with the reviewers' comments: it is important to highlight that the study participants have ALS, and to discuss the relationship between the current findings and humans with normal motor cortex function. (We avoid the use of the word “patient” as the current trial does not medically treat their condition.) To this end, we have modified the Title and Abstract to make this immediately clear, and have expanded the Discussion to consider the implications of these findings for healthy human motor cortex (see text below).

Title: “Neural population dynamics in human motor cortex during movements in people with ALS.”

Abstract: “Here we analyze neuronal ensemble activity in motor cortex in two clinical trial participants diagnosed with Amyotrophic Lateral Sclerosis (ALS). We find that activity in human motor cortex has similar dynamical structure to that of non-human primates, indicating that human motor cortex contains a similar underlying dynamical system for movement generation.”

Discussion: “These results demonstrate prominent rotations of the neural population state in human motor cortex during movements. […] and may also hint at which aspects of motor cortical function are preserved despite the progression of a severe motor neuron disease.”

*3) It will be appropriate to address possible ethical issues of using invasive surgical experiments in patients with a severe disease and limited motor function*.

We agree with the reviewers that ethical issues in human neuroscience are of paramount importance, but we believe that a detailed discussion of these topics is outside the scope of this very brief report focusing specifically on neuronal population dynamics. Nonetheless, we certainly agree that it is important to state the ethical review process that preceded and continues to accompany this research. We have added a sentence to the main text clarifying that this protocol was approved by multiple institutional review boards and is performed under an Investigational Device Exemption granted by FDA. We have also clarified the Methods regarding this point, and to note that these data were not collected solely for this purpose, but were collected as part of a pilot clinical study testing the safety and feasibility of an intracortical neural interface system (BrainGate, www.braingate2.org) for people with tetraplegia. As part of this carefully performed clinical research, research participants contribute enormously by testing evolving algorithms for closed-loop neural cursor control and by providing feedback on the communication and external device control interfaces being developed for use by people with tetraplegia. In addition, the ability to chronically record motor cortical neuronal ensembles in people—for years at a time—provides a simultaneous and unprecedented opportunity to pose questions and to learn about the human nervous system, not only for its fundamental scientific benefits, but directly toward harnessing that knowledge toward the creation of advanced neural interfaces for people with neurologic disease.